# Ultralight Open-Cell Graphene Aerogels with Multiple, Gradient Microstructures for Efficient Microwave Absorption

**DOI:** 10.3390/nano12111896

**Published:** 2022-06-01

**Authors:** Qilin Mei, Han Xiao, Guomin Ding, Huizhi Liu, Chenglong Zhao, Rui Wang, Zhixiong Huang

**Affiliations:** School of Materials Science and Engineering, Wuhan University of Technology, 122 Luoshi Road, Wuhan 430070, China; meiqilin@whut.edu.cn (Q.M.); pxxiaohan@gmail.com (H.X.); ucchecknorth@gmail.com (H.L.); 246012@whut.edu.cn (C.Z.); 303842@whut.edu.cn (R.W.); zhixiongh@mail.whut.edu.cn (Z.H.)

**Keywords:** ultralight graphene aerogels, open cell, gradient microstructure, various microwave absorption

## Abstract

Development of high-performance graphene-based microwave absorbing materials with low density and strong absorption is of great significance to solve the growing electromagnetic pollution. Herein, a controllable open-cell structure is introduced into graphene aerogels by the graphene oxide (GO) Pickering emulsion. The open-cell graphene aerogel (OCGA) with multiple microstructures shows a significantly enhanced microwave absorption ability without any additions. A high microwave absorption performance with the minimum value of reflection loss (RL_min_) of −51.22 dB was achieved, while the material density was only 4.81 mg/cm^3^. Moreover, by means of centrifugation, the graphene cells were arranged by their diameter, and a gradient, open-cell graphene structure was first fabricated. Based on this unique structure, an amazing microwave absorption value of −62.58 dB was reached on a condition of ultra-low graphene content of 0.53 wt%. In our opinion, such excellent microwave absorption performance results from multiple reflection and well-matched impedance brought by the open-cell and gradient structure, respectively. In addition, the structural strength of the OCGA is greatly improved with a maximum increase of 167% due to the introduction of cell structure. Therefore, the OCGAs with the gradient structure can be an excellent candidate for lightweight, efficient microwave absorption materials.

## 1. Introduction

In recent years, emerging technologies such as Internet of Things, 5G communication and satellite networks bring about kinds of electromagnetic waves in our environment, which are harmful to human health, electronic devices, aerospace, information security, etc. Therefore, in order to solve above problems, the development for high-performance electromagnetic absorbing materials with low reflection, wide absorption band and small density becomes an urgent need. Graphene aerogels have been widely studied in the electromagnetic absorption applications because of their excellent electrical properties, lightness and high specific surface area. However, their poor matching impedance and single attenuation mechanism lead to unsatisfactory absorption performance [1]. So various inorganic second phases, such as ferrite [2,3,4,5], Ni [6,7,8], MXene [9,10], SiC [11] and others [12,13] are added in graphene aerogels to tune impedance matching properties for reaching good microwave absorption performance [6]. However, additional inorganic phases usually have a large density, which will increase the weight of aerogels and make the preparation process complex. Thus, the lightweight, pure graphene aerogel realizing efficient microwave absorption has been rarely reported to date [14].

In order to meet the demands for light weight and high efficiency, researchers focus on microstructural adjustment of graphene aerogels. Various methods including mechanical compression, oriented structure and changing porosity have been applied to improve microwave absorption performance of aerogels [4,15,16,17,18]. Reflection loss (RL) is one of the important indexes to evaluate the microwave absorption performance. It refers to the ratio of the incident power of microwave to the reflected power and is usually expressed in dB. For a single-layer microwave absorber, the reflection loss can be expressed as a function of the normalized input impedance of a metal-backed absorber [19,20]. From the relative permeability and dielectric constant in a given frequency range, the reflection loss of the material under different thickness can be calculated (the thickness of the absorber is matching thickness). Although RL_min_ of graphene aerogels can be reduced to below −60 dB by above methods, a large thickness (about 5 mm) or high fill rate (0.74 wt%) is necessary [15,21]. This is mainly due to the single structure feature of normal graphene aerogels, which introduces the single microwave absorption way. Therefore, building multiple microstructures to bring about various absorption mechanisms will be useful for efficient microwave absorption.

Herein, to obtain the strong microwave absorption performance, an open-cell structure was introduced into graphene aerogels via the self-assembly and freeze-drying process. The OCGA with multiple microstructures exhibited an enhanced RL_min_ of −51.22 dB, whose density was only 4.81 mg/cm^3^. To improve the microwave absorption properties further, the OCGA with a gradient cell diameter structure (OCGA-G) was designed and prepared. This gradient aerogels without any additional phase showed a maximum effective absorption bandwidth (EAB, below −10 dB) of 8.55 GHz and a RL_min_ of −62.58 dB at the thickness of 3.2 mm and ultra-low fill rate (0.53 wt%). In our opinion, the multiple, gradient structure of OCGA-G greatly increases the microwave transmission path, strengthens multiple reflection loss and provides an excellent impedance matching characteristic, so that OCGA-G achieves the ultra-light, broadband and strong microwave absorption.

## 2. Materials and Methods

### 2.1. Materials

Pristine graphite (99.95% metals basis, 5000 mesh) and diphenyl ether (DE, 99.0%) were obtained from Shanghai Aladdin Bio-Chem Technology Co., Ltd. (Shanghai, China). Concentrated sulfuric acid (98%), hydrogen peroxide (30%) and Potassium permanganate were obtained from J&K Scientific. N_2_H_4_·H_2_O (80%) was obtained from Yiweian Chemical Technology (Jinan, China). Sliced paraffin (54~56 °C) was obtained from Shanghai hushi chemical reagent and analysis instrument Co., Ltd. (Shanghai, China).

### 2.2. Controllable Preparation of OCGA

GO solution was prepared from graphite powder through an improved Hummer’s method as reported in our previous paper [22,23]. A mixture of DE and GO solution was stirred for 20 min using an Ultra-Turrax in a 60 °C water bath and a series of Pickering emulsions were obtained at different rotational speeds of 8k rpm, 12k rpm, 16k rpm and 20k rpm. These emulsions were placed in ice water mixture for 1 h to freeze the DE emulsion droplets. Then, the emulsions were freeze-dried for 36 h, which would remove the water and a small amount of DE. Next, place the sample in a vacuum oven at 80 °C for 2 h, and the open-cell GO aerogel would be obtained. After that, GO aerogels were reduced by N_2_H_4_·H_2_O vapor at 100 °C for 18 h. Finally, the OCGAs were obtained by heating at 110 °C for 2 h in a vacuum oven to eliminate the residual N_2_H_4_·H_2_O.

### 2.3. Preparation of OCGA-G

The Pickering emulsions prepared at different rotation speeds were mixed together in equal proportion and centrifuged at 2000 rpm for 5 min. The centrifuged emulsion was extracted carefully and frozen dried to obtain gradient GO aerogels. After the reduction treatment in the same way of OCGA, the OCGA-G was prepared.

### 2.4. Characterization

Scanning electron microscopic (SEM) images were taken on a Zeiss Ultra Plus (Oberkochen, Germany) field emission microscope (5.00 kV. FT-IR spectroscopy (Nexus, Madison, WI, USA) was employed to analyze the of functional groups and GO and graphite pellets were prepared using KBr and the samples were scanned in the range from 400 cm^−1^ to 4000 cm^−1^ to obtain the FTIR spectra. Raman spectroscopy was carried out with an inVia confocal Raman microscope (Renishaw, London, UK) using the 514.5 nm laser excitation with an objective magnification of 50× (10 mV spot diameter: 1 μm). The X-ray diffraction (XRD, D/max-rB) patterns were recorded with a scanning rate of 1° per minute in a 2θ range from 5° to 80° with Cu Kα radiation (λ = 1.5418 Å) to characterize the inter layer spacing (Rigaku Corporation, Akishima, Japan). The compression properties of aerogels were tested by RGM-4100 type microcomputer controlled universal material testing machine produced by Shenzhen Regal Instrument Co., Ltd. (Shenzhen, China).

### 2.5. Electromagnetic Parameters Measurement and Microwave Absorption Calculation

The principle of transmission line calculates the input impedance and reflection loss of single-layer microwave absorbing material with a metal backplate. As shown in the figure below, when an incident wave hits on the surface of absorber, it is divided into two parts, one part reflects from the surface into air (R_1_), and the other part infiltrates into the absorber (Figure 1a). The absorber will absorb the entered microwaves and convert them into heat. The electromagnetic wave entering the material is reflected again at the interface between the absorber and the metal plate (R_2_). If R_1_ and R_2_ meet the conditions of interference cancellation, then extinction effect can be obtained, leading to a dissipation of them at the air-absorber interface and resulting in enhanced microwave absorption performance. At this time, the reflected microwave energy will reach the lowest value and the reflection loss reach the peak.

The reflection loss of the material under different thickness can be calculated from the relative permeability and dielectric constant in a given frequency range. The relative complex permittivity (*ε_r_* = *ε*′ − *jε*″) and relative complex permeability (*μ_r_* = *μ*′ − *jμ*″) of OCGAs were measured on a vector network analyzer (Agilent N5224B PNA, Agilent, Santa Clara, CA, USA) at 2–18 GHz (Figure 1b). The test samples were fabricated using melting paraffin to impregnate the OCGAs and cut into standard coaxial rings with an outer diameter of 7.0 mm; an inner diameter of 3.04 mm and a thickness of 4.0 mm. To investigate the microwave absorption properties of aerogels; the reflection loss of the incident electromagnetic wave was calculated according to the following equations [19,20]:(1)$$RL\left(\mathrm{dB}\right)=20lg\left|\frac{Z_{in}-1}{Z_{in}+1}\right|$$
(2)Zin=μrεrtanh(j2πfdcμrεr)
where *RL* is the value of reflection loss, *Z_in_* is the normalized input impedance, *μ_r_* and *ε_r_* are the measured relative complex permeability and permittivity, respectively, *c* is the light speed in vacuum, *f* is the frequency of microwaves and *d* is the thickness of the absorber.

The enhancement of the complex permittivities can be interpreted by Debye relaxation theory as expressed in the following equations [24]:(3)$$\varepsilon^{\prime }\left(\omega \right)=\varepsilon_{\infty }+\frac{\varepsilon_{s}-\varepsilon_{\infty }}{1+\omega^{2}\tau^{2}}$$
(4)$$\varepsilon^{\prime \prime }\left(\omega \right)=\frac{\varepsilon_{s}-\varepsilon_{\infty }}{1+\omega^{2}\tau^{2}}+\frac{\sigma }{\omega \varepsilon_{0}}=\varepsilon_{p}^{\prime \prime }+\varepsilon_{c}^{\prime \prime }$$
where *ω* is the angular frequency, *τ* is the polarization relaxation time, *ε_s_* is the static permittivity, *ε*_∞_ is the relative permittivity in the high-frequency limit, *σ* is the conductivity, *ε*_0_ is the permittivity in vacuum (*ε*_0_ = 8.854 × 10^−12^ F/m), *ε_p_*″ is the imaginary part corresponding to the relaxation polarization, and *ε_c_*″ is the imaginary part corresponding to the conductivity loss.

The quarter-wavelength matching model is used in this study, in which the relationship of absorbers thickness (*t_m_*) with peak frequency (*f_m_*) are described by the following equations [25]:(5)$$t_{m}=\frac{nc}{4f_{m}\sqrt{\left|\varepsilon_{r}\right|\left|\mu_{r}\right|}},\;\;\left(n=1,\;3,\;5\right)$$
where |*ε_r_*| and |*μ_r_*| are the modulus of measured *ε_r_* and *μ_r_* at *f_m_*.

## 3. Results and Discussion

### 3.1. Fabrication of OCGAs

Figure 2 shows the detailed fabrication process of OCGA. The liquid DE and GO aqueous solution were stirred into Pickering emulsion. The microcells with a shell-core structure were prepared by the self-assembly of GO nanosheets at interface between DE and water, and the formation mechanism of this process had been explained in our previous paper [23]. Then, the emulsion was freeze-dried into GO aerogels, and the microcells were opened by DE vapor ejection in a vacuum environment [26,27]. Finally, the OCGA was obtained after chemical reduction using N_2_H_4_·H_2_O vapor. Through the above process, a series of OCGAs was prepared by adjusting the concentration of GO solution, the volume ratio of GO solution to DE and the rotational speed.

### 3.2. Testing and Characterization

In order to explore the preparation process of aerogel, we carried out scanning morphology analysis, infrared spectrum analysis, Raman spectrum analysis and XRD analysis. Through SEM analysis, we found that the micro morphology of aerogel gel is a three-dimensional network structure composed of interconnected open-celled microspheres (Figure 3a). FT-IR spectroscopy is a power full technique to characterize the presence of different functional groups. FT-IR spectrum was recorded, and the spectrum of GO obtained confirmed the successful oxidation of the graphite (Figure 3b). Functional groups such as O-H, C-OH, C-O were observed in GO. Due to the carboxyl OH stretching mode, there is a broad peak between 3500 cm^−1^ and 2500 cm^−1^ in the IR spectrum [28,29]. The peak at around 1725 cm^−1^ is attributed to C=O stretch of carboxyl group [30] while 1230 cm^−1^ corresponds to C-OH stretch of alcohol group [29]. The peak at around 1060 cm^−1^ is attributed to C-O stretching vibrations of C-O-C [31] and the peak around 1620 cm^−1^ is attributed to C=C stretches from unoxidized graphitic domain [28]. All the intensities of the peaks corresponding to the oxygen-containing functional groups of reduced graphene oxide were reduced compared to the intensity of the peaks of graphene oxide, and even the intensities of some peaks disappeared [32]. This suggests that N_2_H_4_·H_2_O successfully reduced graphene oxide. However, all peaks did not disappear, which indicates that the GO was not completely reduced and suggests the presence of some functional groups.

Raman spectra has long been considered as the simplest and non-invasive tool to characterize the structure of carbon [33]. Figure 3c displays the Raman spectra of graphene aerogels with strong D-bands and G-bands centered at 1328 and 1593 cm^−1^, respectively. Additionally, the intensity ratio of the two peaks (D and G) in the Raman spectrum of graphene is higher than that of graphene oxide [34,35], indicating that the number of sp^2^ carbon atoms in graphene is higher than that of sp^3^ carbon atoms, which means that the average size of the sp hybridized carbon planes in graphene is larger than that of graphene oxide. This shows that when graphene oxide is reduced under the present experimental conditions, only part of its sp^3^ carbon atoms are reduced to sp^2^ carbon atoms.

Figure 3d shows the XRD patterns of GO before and after reduction. The GO exhibits a peak at 2θ = 11.1° which corresponds to the reflection of the plane (001) with a corresponding layer to layer distance of 0.796 nm, which is close to in the literature [36]. After reduction of GO, the diffraction peak for graphene oxide was at around 2θ = 25.1°, which corresponds to the reflection of the plane (002), corresponding to a layer-to-layer distance of 0.356 nm [37]. This indicates that the inter layer distance was decreased due to the removal of oxygen containing functional groups when GO is reduced to reduced graphene. Through the above analysis, we confirmed that graphene oxide was successfully reduced to graphene.

### 3.3. Effect of Graphene Framework on the Microwave Absorption Performance

It is well known that the density of graphene aerogels could affect the framework character, which is the usual way to adjust microwave absorption properties. The relationship between the microwave absorption performance of OCGA and its density was investigated at first. OCGAs with densities of 2.43 mg/cm^3^, 4.81 mg/cm^3^ and 7.22 mg/cm^3^ were prepared using the GO solution with the concentration of 5 mg/mL, 10 mg/mL and 15 mg/mL, respectively. The volume ratio of GO solution to DE was 4:1, and the rotational speed was 15k rpm. The obtained OCGA with low, medium and high densities were named OCGA-L, OCGA-M and OCGA-H, respectively.

The framework morphology of OCGAs with different density was observed using SEM. As shown in Figure 4a–c, with the increase of density, the OCGA becomes denser, and the opening rate of the cell decreases obviously. Moreover, there are more graphene lamellae connecting the individual cells together as increase of density. These changes are easy to understand because of the increased graphene content. The different microstructures resulted in different microwave absorption performance of OCGAs, which were measured using the coaxial wire method. The results indicate that all the OCGAs display a typical frequency dependence of permittivity that both *ε*′ and *ε*″ decrease with the increasing frequency. As the graphene content increases, a significant enhancement is achieved in *ε*′ and *ε*″ (Figure 4d,e), which indicates that the aerogels have a growing ability of electrons store and dielectric losses. The change of *ε*′ and *ε*″ is explained as follow. According to Equation (3), *ε*′ is related to the amount of polarization, which represents the ability to store electrical energy from the electric field. Aerogels containing much graphene can produce more dipole-like orientational polarization and interfacial polarization. Therefore, the increase in density of OCGA is responsible for the enhancement of *ε*′. In addition, according to Equation (4), the increase in *ε*″ is not only attributed to enhanced polarization loss, but also to conduction loss. As the content of graphene increases, the conduction of aerogel. This promotes more electron migration between graphene cells and leads to an increase of the conduction loss. Therefore, *ε*′ and *ε*″ both significantly enhance with the increasing graphene content. Furthermore, because the OCGAs are composed of pure graphene without any magnetic component, all samples show *μ*′ and *μ*″ at around 1 and 0, indicating a negligible magnetic loss for incident electromagnetic wave [38].

The microwave absorption performance was calculated and shown in Figure 4f–h. The OCGAs with the lowest or highest density did not show the best microwave absorption ability. This is because that the dielectric loss of OCGA-L is too poor, so a larger thickness is required to reach the RL_min_. For OCGA-H with higher dielectric loss, most microwaves cannot enter inside this dense aerogel due to the mismatched impedance. Proper dielectric loss and good impedance matching (Appendix A) are obtained in OCGA-M, which reaches a RL_min_ of −37.62 dB at the matched thickness of 4.5 mm and the largest EAB of 7.07 GHz at the matching thickness of 4.0 mm. Thus, the OCGA with medium density was adopted in the follow-up study.

### 3.4. Effect of Open-Cell Structure on the Microwave Absorption Performance

Owing to multiple microstructures, the OCGA shows the flexible adjustability. Except for the graphene framework, the cell microstructures, including the number and size of cells, also have dramatic effects on microwave absorption property. The relationship between the cell morphologies and the microwave absorption performance was explored in this study.

#### 3.4.1. Effect of Cell Number on Microwave Absorption Performance

The cell number was regulated by the ratio of DE to GO solution during emulsion preparation process. The OCGAs with a growing number of cells were fabricated using Pickering emulsion with increased DE/GO solution ratios (*v*/*v*), 0, 0.125, 0.250 and 0.375 at the same rotational speed. The obtained graphene aerogels were named GA, OCGA-DE1, OCGA-DE2 and OCGA-DE3, respectively.

As shown in Figure 5a, the GA prepared without DE shows a smooth lamellar structure and has lots of large pores. When a little DE is added, the obtained aerogels still show a lamellar structure, but there are lots of hollow bulges in the lamellae, which are unformed cells with a diameter of about 7 μm (Figure 5b). With the rising content of DE, the number of cells having approximately the same diameter increases further, which disperse uniformly in the aerogel (Figure 5c). However, the graphene lamellae between cells almost disappear when more cells are formed (Figure 5d), because most of the graphene have been assembled to form cells. The electromagnetic parameters of graphene aerogels with different number of cells were tested, and their RL of the aerogels at different matching thicknesses was calculated. The results indicate that the GA without cell structure shows a RL_min_ of −30.65 dB (Figure 5e). When a few cells are formed, the RL_min_ of aerogel increase slightly to −23.34 dB of OCGA-DE1 (Figure 5f). As the number of cells increases further, the RL_min_ decreases first and then increases (Figure 5g,h). Finally, the OCGA-DE2 obtains the smallest RL_min_ of −51.22 dB.

As we all know, a large number of defects and interfaces exist in reduced GO (Figure 5i), leading to various polarization, which can greatly consume the incident microwave. For the conventional graphene aerogels without cells, the graphene framework provides lamellae for electromagnetic wave reflection and conductive paths for electron migration. However, the graphene lamellae occupy less space, and there is little apparent obstacle for the transmission of microwave (Figure 5j). Therefore, there is a little reflection loss in this aerogel. When a small amount of DE is added, the emerging hollow bulges affect the electron migration, which will reduce the conduction loss, so that microwave absorption property decreases firstly. Then with the increase of cell number, the cells are opened gradually, and the specific surface area of the OCGA increases obviously. The various loss mechanisms in this multiple structure are formed, as shown in Figure 5k. On the one hand, beyond absorption of graphene framework, the open-cell structure results in frequent refraction and reflection of microwaves inside and outside the cells, which greatly increases the transmission path of microwave and improves the reflection loss. On the other hand, we consider that the cells construct a microscale spherical conductive structure, which can respond to incidence microwaves as huge resistance-inductance-capacitance coupled circuits [39]. When the incident microwaves reach this structure, the induced currents under time-varying electromagnetic fields are generated easily in the spherical circuit and converts the electromagnetic energy into heat. Based on above theory, the various loss mechanisms for microwaves exist in OCGA with modest cells, which will greatly improve the microwave absorption ability of aerogel. However, when the number of cells continues to increase, the graphene lamella connecting cells disappears gradually, which decrease the transmission of electron and causes absorption ability to decrease again. Overall, the OCGA with moderate cells achieves the best microwave absorption property, and the ratio of DE to GO solution in emulsion preparation process is set to 1:4 in our next study.

#### 3.4.2. Effect of Cell Size on Microwave Absorption Performance

Furthermore, the effect of cell size of OCGA on microwave absorption performance was also studied, which could be adjusted by the rotational speeds when the GO Pickering emulsions were prepared. The OCGAs with different cell sizes were obtained by applying different rotational speeds of 10k rpm, 15k rpm, 20k rpm and 25k rpm, which were named OCGA-10K, OCGA-15K, OCGA-20K and OCGA-25K, respectively. The morphology of OCGAs was observed, and their SEM images were shown in the Figure 6a–d. The graphene cells are regular, opening spheres and the size of the cells decreases from 30 μm in OCGA-10K to 4.5 μm in OCGA-25K, which corresponds to the microphotograph of their Pickering emulsions (Appendix A). So, the cell size can be controlled easily by the rotation speed. The electromagnetic parameters of OCGAs with different cell sizes were measured, and their RL were also calculated. As shown in Figure 6e, the peak location of RL_min_ rises from 8.42 GHz for OCGA-10K to 10.48 GHz for OCGA-15K, 11.38 GHz for OCGA-20K and 17.87 GHz for OCGA-25K in order. In addition, the matching thickness of the RL_min_ is reduced from 5.0 mm for OCGA-10K to 3.0 mm for OCGA-25K (Figure 6f). This is due to the progressive enhancement of the dielectric loss of OCGA to microwaves as the diameter decreases (Appendix A). Furthermore, the RL_min_ of OCGAs decreases at first and then increases, which reach a minimum value of −51.22 dB at OCGA-20K (Figure 6e). Thus, the microwave absorption performance of OCGAs achieves the optimum state of impedance matching characteristics (Appendix A) and attenuate ability when the diameter of cells is about 6 μm.

Moreover, we find an interesting phenomenon that when the thickness of OCGAs is fixed, the change of cells size has very little effect on the peak location of RL, which only changes peak strength (Figure 6g). The regular methods of adjusting microwave absorption performance, such as changing the composition or deformation of materials, usually affect both the peak location and strength of RL, which can hardly be controlled individually. In our opinion, this property of OCGA is due to the constant volume fraction of graphene in the different OCGAs, causing an almost indistinguishable modulus of complex permittivity (Figure 6h). Owing to fixed matching thickness, constant modulus of the complex permittivity and permeability, the change of cell size has little effect on the peak location according to Equation (5). This characteristic greatly expands the application of OCGA. For example, if an absorber for 11 GHz to 12 GHz is demanded, the RL_min_ of our material can be adjusted freely from about −23 dB to −45 dB, and meanwhile there is little interference to other frequency bands.

The compressive strength of graphene aerogels is also obviously improved when the cell structure is introduced. The compression performance of GA and OCGAs with different cell sizes was tested. The results show that compression strength of GA without cell structure is 3.6 kPa. When the cell structure exists, the compressive strength of OCGAs increases obviously, and its maximum value reaches 9.5 kPa of OCGA-20K with a 167% increase (Figure 7a). Based on this, the OCGA-20K can withstand a 1.6 × 10^4^ times heavier weight than its own mass. To visualize the increase of compressive strength, a counterweight (200 g) was placed on different graphene aerogel. As shown in Figure 7b, the deformation of the normal GA is large, and its shape is difficult to recover after removing the weight. By contrast, the deformation of the OCGA-20K is smaller after adding same weight, and it almost fully recover after removing the weight (Figure 7c). It should be noted that these two aerogels contain the same content of graphene. We consider that the increase of compressive properties of OCGAs is mainly due to the spherical cell structure that is supposed to bear much pressure.

### 3.5. OCGA with Gradient Structure for Efficient Microwave Absorption

Gradient design is acknowledged as an effective strategy for improving the microwave absorption ability nowadays [40]. In order to reduce the thickness and improve the absorption performance of OCGAs further, the size distribution of cells is adjusted to obtain better impedance matching characteristics. By means of density difference between DE (1.08 g/cm^3^) and GO solution (1.01 g/cm^3^), the DE microspheres in Pickering emulsion can be accelerated to settle by the simple centrifugation treatment (Figure 8a). In addition, based on the equation of Stokes law (Equation (6)) [41], the larger the microspheres are, the greater the sedimentation rate is.
(6)$$v=\frac{2gr^{2}\left(\rho -\rho^{\prime }\right)}{9\eta }$$

Here, *g* is the acceleration due to gravity, *r* is the droplet radius, *ρ* is the density of the dispersed phase (DE in this study), *ρ*′ is the density of the continuous phase (GO aqueous solution in this study), and *η* is the shear viscosity of the continuous phase. Based on this, the Pickering emulsions containing DE microspheres with different sizes were centrifuged to obtain a gradient distribution. The OCGA-G with gradient cell size cell was first prepared using this centrifugated emulsion through the process of freeze-drying, vapor ejection and chemical reduction in turn. Furthermore, as comparison groups, the OCGAs using the same Pickering emulsions after mixing (OCGA-m) and precipitating for 2 h (OCGA-P) were also prepared, respectively.

The SEM of OCGA-G shows an obvious gradient structure along thickness direction (Figure 8b–d). At the top of OCGA-G, the cells are relatively integrated with an average diameter of 7 μm and packed tightly. The middle position mainly contains the medium opening cells with a large average diameter of 12 μm. The bottom region of OCGA-G is very loose and has the largest, completely opening cells with an average diameter of 28 μm. These structure characteristics are due to that at the top regions, the small cells bear light centrifugal force, so that its structure is nearly unbroken. But the large microspheres at bottom bear strong centrifugal force, which causes emulsion breaking easily during the centrifugation. By comparison, OCGA-m does not show the gradient structure (Appendix A), and OCGA-P has an inconspicuous gradual morphology (Appendix A).

The distribution state of the cells has a significant effect on the electromagnetic parameters (Appendix A) and microwave absorption performance of OCGAs. The RL_min_ for OCGA-m (Figure 8e) is −35.44 dB at a matching thickness of 5 mm. After precipitation treatment, the matched thickness of OCGA-P decreases to 3.4 mm, the RL_min_ decreases slightly to −42.88 dB. Furthermore, the peak shifts towards the high frequency region (Figure 8f). For OCGA-G, the RL_min_ is declined dramatically to −62.58 dB at 14.73 GHz, and the matching thickness is also reduced to 3.2 mm. Furthermore, the maximum value of EAB reaches 8.55 GHz in OCGA-G at a matching thickness of 3.7 mm (Figure 8g,h). It is noteworthy that the graphene filling rate in OCGA-G is only 0.53%, which is apparently lower than that of graphene aerogel with similar microwave absorption performance (graphene filling rate is 0.74 wt%) in the reported literature [15].

We consider that these outstanding performances of OCGA-G root in its gradient structure, and the different positions in aerogels play different roles in the microwave absorption. Firstly, the large opening cells at the bottom of OCGA-G can improve the impedance matching between the surface and the air and reduce the reflection of microwave, which make large amounts of microwave enter the aerogels (Appendix A). Then, with the transmission of microwave into interior OCGA-G, the smaller, packed graphene cells provide more reflection interfaces, which gradually increase dielectric loss according to previous multiple absorption mechanism. To sum up, by means of good gradient impedance and enhanced microwave loss capability, the ultralight OCGA-G shows a lower value of RL and smaller matched thickness than the others.

## 4. Conclusions

In summary, OCGAs with open-cell structures were prepared using the GO Pickering emulsion, which is a meaningful way to fabricate large-size aerogels for practical applications. The cell structures including number and size could be changed by the ratio of DE to GO solution and rotational speed, respectively. By means of the changeable multiple microstructures, OCGAs had adjustable impedance matching and attenuation characteristics, which will greatly optimize microwave absorption properties without any additional phase. Moreover, a gradient design for diameter distribution of cells was first described, and the OCGA-G with gradient structure showed surprising electromagnetic performance that RL_min_ reached −62.58 dB at a thickness of only 3.2 mm, and the EAB was as wide as 8.55 GHz. In addition, the structural strength of the OCGA was greatly improved with a maximum increase of 167% due to the introduction of cell structure. Thus, this work provides an inspiration for the design and preparation of ultralight, efficient microwave absorption materials using multiple structures.

## Figures and Tables

**Figure 1 nanomaterials-12-01896-f001:**
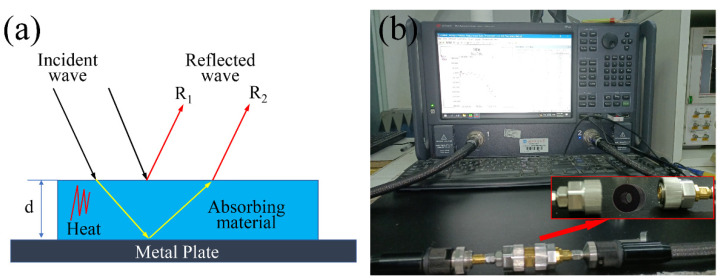
(**a**) The schematic diagram of microwave absorption material. (**b**) Schematic diagram of vector network analyzer and coaxial sample.

**Figure 2 nanomaterials-12-01896-f002:**
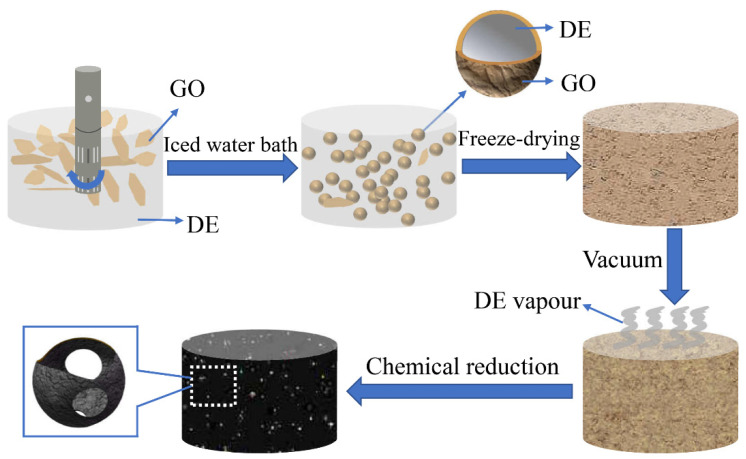
Schematic illustrations of the fabrication of the OCGA.

**Figure 3 nanomaterials-12-01896-f003:**
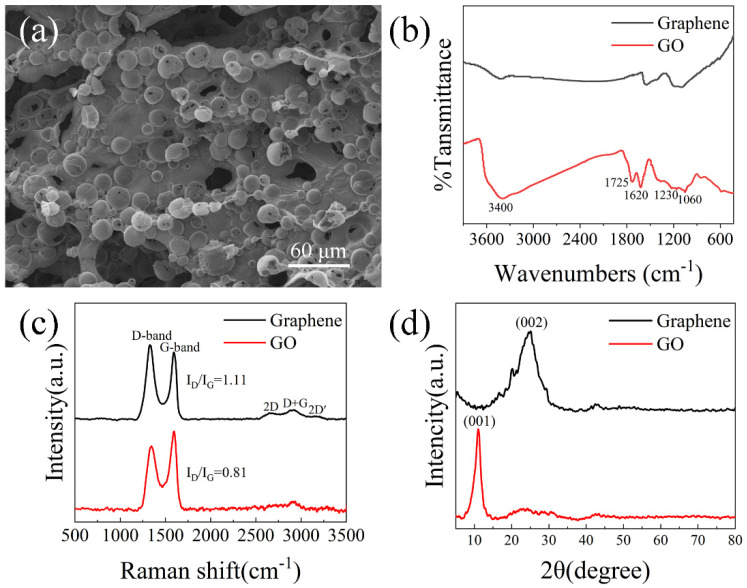
Characterization of OCGA. (**a**) SEM image of OCGA. (**b**) FTIR spectroscopy, (**c**) Raman spectra and (**d**) XRD patterns of GO before and after reduction.

**Figure 4 nanomaterials-12-01896-f004:**
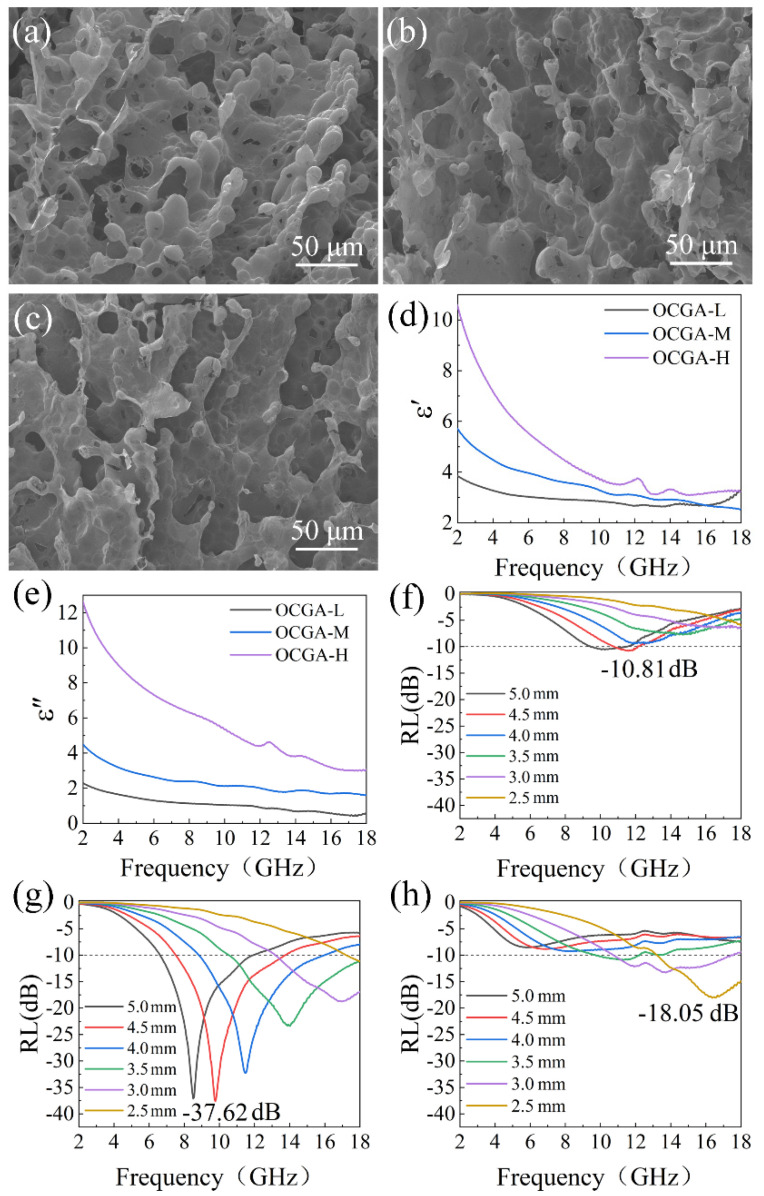
Characterization of OCGAs with different graphene framework. SEM images of (**a**) OCGA-L, (**b**) OCGA-M and (**c**) OCGA-H. (**d**) Real part (*ε*′) and (**e**) imaginary part (*ε*″) of permittivity, and RL of (**f**) OCGA-L, (**g**) OCGA-M and (**h**) OCGA-H.

**Figure 5 nanomaterials-12-01896-f005:**
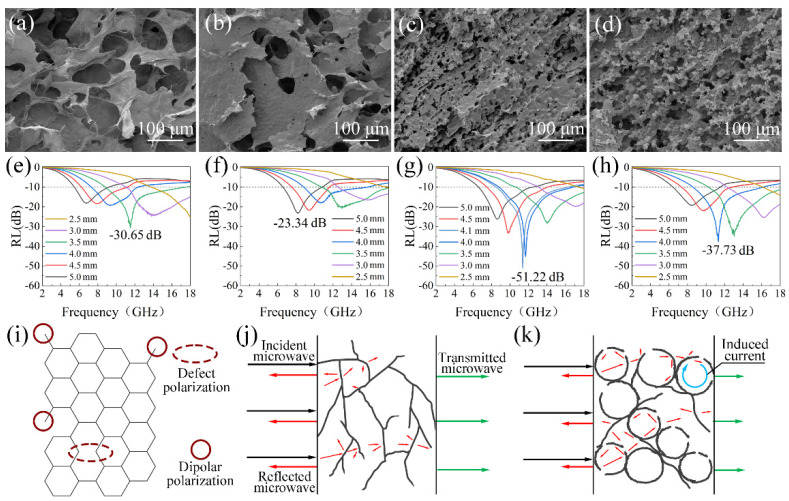
Effect of cell structure on the microwave absorption performance. SEM images of (**a**) GA, (**b**) OCGA-DE1, (**c**) OCGA-DE2, (**d**) OCGA-DE3. RL of (**e**) GA, (**f**) OCGA-DE1, (**g**) OCGA-DE2 and (**h**) OCGA-DE3. (**i**) Polarization from carbon. (**j**) Microwave absorption mechanism of GA (The black arrow indicates incident microwave, the red arrow indicates reflected microwave, and the green arrow indicates transmitted microwave). (**k**) Microwave absorption mechanism of OCGA.

**Figure 6 nanomaterials-12-01896-f006:**
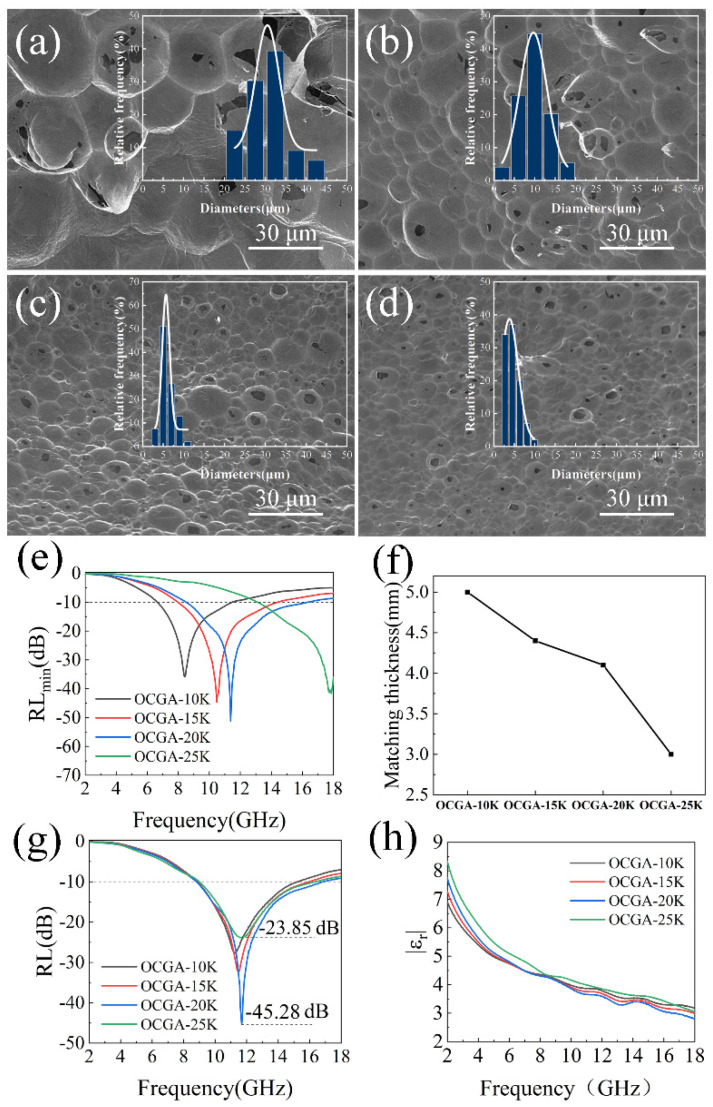
Characterization of OCGA with different cell diameters. SEM images of (**a**) OCGA-10K, (**b**) OCGA-15K, (**c**) OCGA-20K and (**d**) OCGA-25K, showing an average diameter of about 30 μm, 9 μm, 6 μm and 4.5 μm, respectively. (**e**) RL_min_ of the OCGAs, reaching −35.89 dB, −44.65 dB, −51.22 dB and −41.61 dB, respectively. (**f**) Matching thickness of OCGA at RL_min_. (**g**) RL of OCGA at the matching thickness of 4 mm. (**h**) Modulus of the complex permittivity of OCGA.

**Figure 7 nanomaterials-12-01896-f007:**
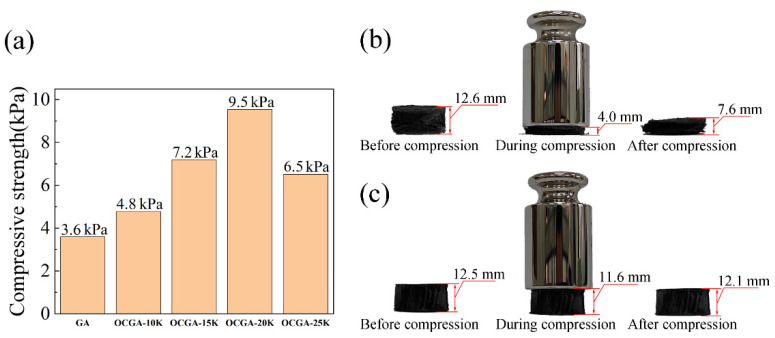
Effect of cell structure on the compression performance. (**a**) Compressive strength of graphene aerogels. (**b**) Shapes of GA before, during compression and after compression. (**c**) Shapes of OCGA-20K before, during and after compression.

**Figure 8 nanomaterials-12-01896-f008:**
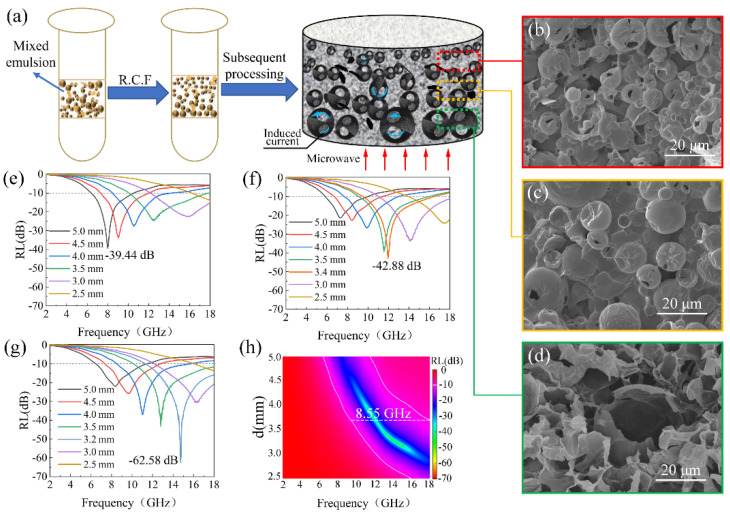
Effect of cell distribution on the microwave absorption performance. (**a**) Schematic description of OCGA-G. SEM images of (**b**) top region, (**c**) middle region and (**d**) bottom region of OCGA-G. (**e**) RL of OCGA-m. (**f**) RL of OCGA-P. (**g**) RL and (**h**) EAB of OCGA.

## Data Availability

Not available.

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
