# Peer review of "Ultralight Open-Cell Graphene Aerogels with Multiple, Gradient Microstructures for Efficient Microwave Absorption"

_nanomaterials, 2022, doi:10.3390/nano12111896_

Round 1

Reviewer 1 Report

Dear authors, 

 the manuscript is discussing the fabrication, characterization and performances of open-cell graphene aerogels for microwave absorption and it also a certain degree of novelty related to the use of pure aerogel with changeable multiple microstructure. 

However, in the paragraph 2.1 "characterization", three techniques for the characterization of the microstructure are reported, namely SEM, metallographic microscopy and Raman spectroscopy, but only SEM results are the presented and discussed.

In addition, for none of them sufficient information are provided (e.g.: for SEM images: kV; for Raman spectroscopy: magnification of the objective, power, numerical aperture, etc).

Therefore, I’d suggest to revise the manuscript including the results of the characterization with all the techniques mentioned in the text. In particular Raman spectroscopy is used to characterize the structural changes before and after the reduction of GO, thus providing important information about the graphene aerogels prepared using the GO Pickering emulsion, for instance about the purity of the graphene material, the level of disorder, etc.

The SEM images alone don’t provide sufficient information about the quality of the material, and therefor e I’d recommend the publication of the manuscript after the integration and discussion with the chemical/physical characterization of the material.

Author Response

Response to Reviewer 1 Comments

Point 1: The manuscript is discussing the fabrication, characterization and performances of open-cell graphene aerogels for microwave absorption and it also a certain degree of novelty related to the use of pure aerogel with changeable multiple microstructure.

However, in the paragraph 2.1 "characterization", three techniques for the characterization of the microstructure are reported, namely SEM, metallographic microscopy and Raman spectroscopy, but only SEM results are the presented and discussed. In addition, for none of them sufficient information are provided (e.g.: for SEM images: kV; for Raman spectroscopy: magnification of the objective, power, numerical aperture, etc).

Response 1: We deleted the metallographic microscope test because it only appears in Electronic Supplementary Information. The power of SEM test is 5.00 kV. As for Raman spectroscopy, its magnification of the objective is 50× and the power is 10 mV. As the test instrument is Nanobase Raman spectrometer, which uses slit-CCD confocal mode, there is no numerical aperture. Its slit width is 65 μm and spot diameter is 1 μm. Relevant information has been added to the manuscript.

Point 2: Therefore, I’d suggest to revise the manuscript including the results of the characterization with all the techniques mentioned in the text. In particular Raman spectroscopy is used to characterize the structural changes before and after the reduction of GO, thus providing important information about the graphene aerogels prepared using the GO Pickering emulsion, for instance about the purity of the graphene material, the level of disorder, etc.The SEM images alone don’t provide sufficient information about the quality of the material, and therefor e I’d recommend the publication of the manuscript after the integration and discussion with the chemical/physical characterization of the material.

Response 2: At your suggestion, we analyzed the Raman spectra before and after the reduction of aerogels. In addition, the chemical/physical characterization of the material was characterized by FT-IR/XRD analysis before and after the reduction of aerogel. Relevant information is reflected in Section 3.2 of the responsed manuscript.

Reviewer 2 Report

Dear Editor,

This article presents interesting research on the microwave absorption properties of graphene-based aerogels. My major concern about this article is that, although the motivation of the research has high societal impact, it is written towards a more expert audience that the average readership of this journal. Rather than recommending the authors to submit the article to a specialized journal, I think that it can well fit in Nanomaterials, provided the authors invest in enhancing the introductory part and measurement setup sections of the manuscript. Below I give some suggestions.

The article uses extensively the concept of “reflection loss (RL)” and “matching thickness.” I recommend the authors to expand on these concepts including schematic diagrams and actual pictures of their measurement setup. The only partial description of the setup is that aerogels were cut into standard coaxial rings. What is the matching thickness in this scheme? Is it the length of the coaxial “cable” made by the aerogel? Also, how is absorption related to RL? Of course, less reflection means more absorption, but it also means more transmission. Can the authors explain how their setup rules out transmission (or radiation from the aerogel) and attributes less reflection solely to more absorption?

In addition, I found some minor typo-like error in the manuscript that I pass to the authors:

- In line 245, it should say Figure 4 instead of 3.

- Consider relabeling the y-axis in Fig. 4e to RLmin to make it easier to understand that each curve was taken at a different matching distance.

- Line 306 mentions “placing” and line 325 mentions “precipitation.” Are these equivalent terms? If not, or even if they are, consider using just one to avoid confusions.

Author Response

Response to Reviewer 2 Comments

Point 1: This article presents interesting research on the microwave absorption properties of graphene-based aerogels. My major concern about this article is that, although the motivation of the research has high societal impact, it is written towards a more expert audience that the average readership of this journal. Rather than recommending the authors to submit the article to a specialized journal, I think that it can well fit in Nanomaterials, provided the authors invest in enhancing the introductory part and measurement setup sections of the manuscript. Below I give some suggestions.

The article uses extensively the concept of “reflection loss (RL)” and “matching thickness.” I recommend the authors to expand on these concepts including schematic diagrams and actual pictures of their measurement setup.

Response 1: Reflection loss (RL) is one of the important indexes to evaluate the microwave absorption performance. It refers to the ratio of the incident power of microwave to the reflected power (Pref/Pin) and is usually expressed in dB. For a single-layer microwave absorber, the reflection loss can be expressed as a function of the normalized input impedance of a metal-backed absorber. From the relative permeability and dielectric constant in a given frequency range, the reflection loss of the material under different thickness can be calculated (the thickness of the absorber is matching thickness). Figure (a) above shows the schematic diagram of microwave absorption material, and Figure (b) Schematic diagram of vector net-work analyzer and coaxial sample.

Point 2: The only partial description of the setup is that aerogels were cut into standard coaxial rings. What is the matching thickness in this scheme? Is it the length of the coaxial “cable” made by the aerogel?

Response 2: The matching thickness is 4.0 mm in this scheme, and it is not the thickness of the coaxial cable. We can see the details of the coaxial ring sample test in Figure (b).

Point 3: Also, how is absorption related to RL? Of course, less reflection means more absorption, but it also means more transmission. Can the authors explain how their setup rules out transmission (or radiation from the aerogel) and attributes less reflection solely to more absorption?

Response 3: Reflection loss is usually calculated and reported as a function of the thickness of microwave absorption material. Generally speaking, the smaller the reflection loss, the wider the effective absorption bandwidth (the frequency range in which the reflection loss is less than -10 dB), the better the microwave absorption performance.

The principle of transmission line calculates the input impedance and reflection loss of single -layer microwave absorbing material with a metal backplate. As shown in the figure below, when an incident wave hits on the surface of absorber, it is divided into two parts, one part reflects from the surface into air (R1), and the other part infiltrates into the absorber. The absorber will absorb the entered microwaves and convert them into heat. The electromagnetic wave entering the material is reflected twice at the interface between the absorber and the metal plate (R2). If R1 and R2 meet the conditions of interference cancellation, then extinction effect can be obtained, leading to a dissipation of them at the air-absorber interface and resulting in enhanced electromagnetic wave absorption performance. At this time, the reflected microwave energy will reach the lowest value and the reflection loss reach the peak.

Point 4: In addition, I found some minor typo-like error in the manuscript that I pass to the authors:

- In line 245, it should say Figure 4 instead of 3.

Response 4: Thank you for your advice and we have corrected this error.

Point 5: Consider relabeling the y-axis in Fig. 4e to RLmin to make it easier to understand that each curve was taken at a different matching distance.

Response 5: We have considered your suggestion and made adjustments.

Point 6: Line 306 mentions “placing” and line 325 mentions “precipitation.” Are these equivalent terms? If not, or even if they are, consider using just one to avoid confusions.

Response 6: Yes, they are. And we have made adjustments to avoid confusions.

Round 2

Reviewer 1 Report

Dear authors

thank you for reviewing the manuscript, that I suggest for publication in the present form